
**Singular spectrum and principal component analysis of soil radon**
**(Rn-222) emanation for better detection and correlation of seismic**
**induced anomalies**
Timangshu Chetia[1,2], Saurabh Baruah[1], Chandan Dey[1,2], Sangeeta Sharma[1], Santanu Baruah[1]
[1]Geoscience & Technology Division, CSIR-North East Institute of Science and Technology
(CSIR-NEIST), Jorhat-785006, Assam, India
[2]Academy of Scientific and Innovative Research (CSIR-NEIST), Jorhat-785006, Assam, India
*Correspondence to*: Santanu Baruah (santanub27@gmail.com)
**Abstact.** In the recent years there are several reporting's of anomalous seismic induced
temporal changes in soil radon emanation. It is however well known that radon anomalies apart
from seismic activity are also governed and controlled by meteorological parameters. This is
the major complication which arise for isolating the seismic induced precursory signals. Here
in the investigation the soil radon emanations temporal variability at MPGO, Tezpur, is
scrutinized in the lime light of singular spectrum analysis (SSA). Further prior applying SSA
Digital filter (Butterworth low pass) is applied to remove the high frequency quasi periodic
component in the time series of soil radon emanation. It was scrutinized that sum of just 9
eigenfunctions were sufficient enough for reproducing the prominent characteristics of the
overall variation. This perhaps also evinces that more significantly produced fluctuations are
mostly free from natural variations. The variations in soil temperature was observed to be
dominated by daily variations similar to radon variation which account to 97.99 % whereas soil
pressure accounts for 100 % of the total variance which suggests that daily variations of soil
radon (Rn-222) emanation are controlled by soil pressure in MPGO, Tezpur during the
investigation period followed by soil temperature. The study concludes that SSA eliminates
diurnal and semidiurnal components from time series of soil radon emanation for better
correlation of soil radon emanation with earthquakes.




## 1   Introduction

Radon (Rn-222) is a noble gas, a decay product of radium with atomic number, Z=86
and a half-life of nearly 3.8 days. Because of its short decay time, amount changes in its
production from the rock is quite evidenced. Ulomov and Mavashev in the year 1971 (Ulomov
and Mavashev, 1971) first suggested the correlation of radon concentration with earthquakes.
It has been scrutinized that the radon concentration has correlation to earthquakes and volcanic
eruptions (Walia et al. 2006, Singh et al. 2005). Significant radon concentration anomalies were
also observed prior to the Uttarkashi earthquake of 20th October, 1991; $m_b$~6.5 (Virk and Singh
1994). Radon concentrations was monitored in the North West Himalaya for earthquake
prediction studies and empirical equation between earthquake magnitude, epicentral distance
and precursory time were examined (Kumar et al. 2009). Earthquake prediction depending
entirely on precursory phenomena is empirical and comprises many applied problems. Among
various geophysical parameters soil radon is preferred and used for earthquake precursory
studies because of its ease of detectability. Radon in nature has different isotopes: Rn-222 (half-
life~3.8 days), Rn-220 (Thoron, half-life~54.5 s) and Rn-219 (half-life~3.92 s). The utmost
prominent is Rn-222 because of its longer half-life which is a product of Ra-226 decay process.
The Rn-222 emanates from soil or crust either by diffusion or convection and reaches the
atmosphere. The soil radon emanation concentration is generally assigned to developments of
micro-cracks, fissure and fracture due to dilatancy prior to earthquake. This process enhances
the transportation of Radon from its original enclosure following the cracks into the
atmosphere. Significant radon concentration anomalies were also observed prior to the
Uttarkashi earthquake of 20th October, 1991; mb~6.5 (Virk and Singh, 1994).
North-East India (NE India) is highly vulnerable to earthquake and lies in seismic zone
V (BIS 2002) and frequent occurrence of earthquake facilitates the probability of finding
precursory phenomena which may lead to a successful prediction in near future.  With this

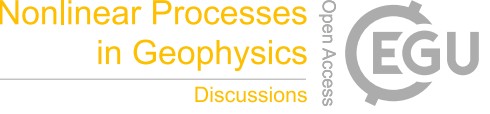

objective a Multiparametric Geophysical Observatory (MPGO) in Ouguri Hills (Latitude
26.61°; Longitude 92.78°, Elevation~82m), Tezpur, Assam, India with the installation of
several geophysical instruments collecting data simultaneously, portray an opportunity towards
identification of precursory signatures prior to earthquakes.    Earthquake precursory and
prediction studies advanced in late 1970s and Heicheng earthquake of 4$^{th}$ February is a land-
mark which was in short-term successfully predicted in 1975 in China (Adams, 1976). The
accomplished medium term forecast of M~7.5 earthquake on 6$^{th}$ August, 1988 in northeast
Indian region (Gupta and Singh, 1988) encouraged to strengthen such studies in India. Another
successful short term prediction was done of (M $\geqslant$ 4) in Koyna region of western India, famous
for Reservoir Triggered Seismicity (Verma and Bansal, 2012).
The study tries to correlate radon emanation in soil gas with earthquakes within the
epicentral distance of 100 km of mb > 3.1 from MPGO, Tezpur which is situated in a highly
tectonically strained and seismically active region. The major problem arises is the removal of
quasi periodic, diurnal (mostly due to temperature) and semidiurnal (mostly due to pressure)
components (Kumar et al., 2015). Radon anomalies are governed by seismic activity as well as
by meteorological parameters (soil temperature, pressure, rainfall, moisture and even wind for
atmospheric radon; Stranden et al., 1984; Kumar et al.,2009;Walia et al., 2005) which in turn
makes it more complex for identifying the seismic induced anomalies. Here in the investigation
characteristics features of temporal soil radon concentrations variability at MPGO, Tezpur is
scrutinized by applying singular spectrum analysis (SSA) in concern to the objective of filtering
the meteorological parameters on radon emanation. SSA is a relatively innovative and powerful
advanced method which has been used across many applied problems for different scientific
fields (e.g., Fraedrich, 1986, Serita et al., 2005). The foremost concept of SSA is applying PCA
on trajectory matrix acquired from the original time series following time series reconstruction.



## 2     Seismotectonics of the region


In middle of the active Kopili and Bomdila fault, the MPGO is situated. The Kopili
and Bomdila faults comprise Neogene-Quaternary sediments, which directly were deposited
over the Archean basement. The Kopili fault zone in an approx is 100 km in width and 300 km
in length. It is a NW-SE trending strike-slip fault (Kayal et al., 2006, Bhattacharya et al., 2008,
2010). The two Precambrian massifs - the Shillong Plateau and the Mikir Hills is delineated by
the tectonic disposition of the Kopili fault. MPGO is bounded to the north by the Main
Boundary Thrust (MBT) and to the south by the NE-SW trending Belt of Schuppen (Figure 1).
The Bomdila fault is strike slip fault of about 400 km in length which trends along WNW-ESE
direction. The northern portion of the fault mostly lies in the Gondwana, Paleogene and
Neogene sediments. This fault is surrounded by the Belt of Schuppen to the east and south by
the Mikir massif and to the west. The fault cuts across the Himalayan fold belt towards the
north (Nandy and Dasgupta, 1991).
The Kopili Fault has produced two large earthquakes (Figure 1) $M_w$~7.7, 1869 event
(Figure 1)  in the south eastern edge of the fault contravening the Naga-Disang thrust and
$M_w$~7.2, 1943 earthquake which occurred farther to the north of 1869 event within a period of
nearly 75 years (Kayal, 2008). It is highly active with strong seismicity discernible down up to
depth of about 50 km, and which extends to the Main Central Thrust (MCT) in the Bhutan
Himalaya. Even if MCT is dormant (Ni and Barazangi, 1984), intense activity is observed at
the region where Kopili Fault meets the MBT and MCT (Nandy, 2001, Kayal et al., 2010,
2012). This is demonstrated by the August 19, 2009 Earthquake ($M_w$~5.1) in the Assam Valley
that occurred in the center of the Kopili fault zone and the September 21, 2009 strong Bhutan
Himalaya Earthquake ($M_w$~6.3) that occurred at the northern end of the Kopili fault where it
connects with the MCT (Kayal et al., 2012). Both the earethquakes are shallow focus (depth ~
10 km) showing right lateral strike-slip faulting (Kayal et al., 2010) which suggests that the





Kopili fault zone is experiencing compressional stress due to the Indo-Burma arc and Himalyan
arc to the east and the north respectively which is characterized by transverse tectonics. The
Bomdila Fault inter-weaves across three major tectonic domains of the NE-India, namely
MCT, MBT and Naga-Disang thrust along NW-SE direction. The earthquake events along
Bomdila fault occur in a diffused pattern having post-collisional intracratonic characteristics
(Nandy and Dasgupta, 1991). It is observed that, the Upper Brahmaputra Valley stretching
between the Bomdila Fault and almost near NW-trending Mishmi Thrust in the northeast is
seismically dormant, and is recognized as the Assam Gap (Khattri, 1983).
**3      Method and techniques**
**3.1     Soil radon (Rn-222) time series**
A BMC2 barasol manufactured by Algade is into operation for soil radon emanation
time series data in MPGO for earthquake prediction and precursory studies. Soil gas radon
emanation every 15 minutes is being continuously monitored. The barasol probe is kept fitted
inside a plastic tube (length 1.5m and diameter of 0.0635m) with open bottom dug inside the
ground in such a way that the detection unit (detector sensitivity-0.02 pulses/h for 1 Bq/m$^3$ and
saturation volumetric activity-3MBq/ m$^3$) which is at the bottom of the probe lies 1m from the
ground level. A silicon alpha detector detects the radon gas which enters the detection chamber
when it emanates from the soil. The radon pass in a detection volume over three cellulose filters
which allows to trap all the solid daughter products of radon. The sensor is a fixed silicon
detector with a depleted depth of 100μm and 400 mm$^2$ of sensitive area. It performs the
counting by atomic spectrometry of radon (Rn-222) and daughter products created in the
detection volume (with window customized between 1.5 MeV and 6 MeV). The probe system
is embedded with soil pressure and temperature sensor. The sensor calibration permits the
volumic activity of the radon (Rn-222) to be evaluated. In the present investigation the soil



radon emanation temporal variability at MPGO, Tezpur the radon data were prudently checked
for no gaps or discontinuous jump. Digital filter (Butterworth) is applied to eliminate the high
frequency quasi periodic components form the soil radon time series for better discernibility of
seismic induced anomalies.
**3.2     Singular Spectrum Analysis**
The SSA results and graphs in the investigation are acquired by using Caterpillar-SSA
3.40 software (Alexandrov and Golyandina, 2004). Window selection rule applied to the time
series is one half of the length of the time series to meet the theoretical requirements for the
investigation (Golyandina, 2010, Khan, 2011, Hassani, 2007). The singular value
decomposition (SVD) alogorithm was applied as it is more accurate than QR iteration which
are the common most algorithm for solving of eigenvalues and singular value problems
(Demmel, 1992).  The main objective of SSA is decomposing the original time series into sum
of series such that each of the component in this sum can be known (either as a trend, periodic
or quasi-periodic components) or noise. This is accomplished by decomposition and
reconstruction. At the first the time series is decomposed following the reconstruction of the
original time series (which is without noise). The methodology adopted here is first to
embedding a 1-dimension time series say, $Y_T = (y_1,...,y_T)$ into a multi-dimensional time series
$X_1,...,X_K$ having vectors $X_i = (y_i,...,y_{i+L-1}) \in R^L$ (Golyandina et al., 2001, 2001). Here the value
of $K = T - L + 1$. The $X_i$ Vectors are called $L$-lagged vectors. The embedding depends on the
window length $L$, such that $2 \leq L \leq T$ which results for trajectory matrix (Hankel matrix:
diagonal elements $i + j$ = const. are equal) $X = [X_1,...,X_K] = (X_{ij})\,_{i,j=1}^{L,K}$. Secondly the Singular
Value Decomposition (SVD) of the trajectory matrix is performed to represent it as an addition
of bi-orthogonal elementary matrices having rank one. Represented by $\lambda_1,...,\lambda_L$ which are the
Eigen-values of $XX'$ in a descending order of magnitude ($\lambda_1 \geq ...\lambda_L \geq 0$) and by $U_1,...,U_L$ which



are the orthonormal system (i.e. $U_i$, $U_j$ =0 for $i \neq j$) is the orthogonality property) and $\|U_i\|$=1,
of the eigenvectors of the matrix $XX'$ corresponding to these eigenvalues. ($U_i$, $U_j$ ) is the inner
product of the vectors $U_i$ and $U_j$ and $\|U_i\|$is the norm of the vector $U_i$. The Set
$$d = \max (i, \text{ such that } \lambda_i > 0) = rank\ X \qquad \text{(I)}$$

If we represent $V_i = X'\ U_i/ \sqrt{\lambda_i}$, then SVD of the trajectory matrix can be represented
as:
$$X = X_1 + \cdots + X_d \qquad \text{(II)}$$

Here $X_i = \sqrt{\lambda_i}U_iV_i'$ ($i = 1,...,d$).
Thirdly the series reconstruction is accomplished by grouping, to split the elementary
matrices ($X_i$) into various groups and addition of the matrices within each and every group. Say
$I = \{i_1,...,i_p\}$ be group of indices $i_1,...,i_p$. Then $X_I$ matrix parallel to the group $I$ is defined as $X_I$
$= X_{i1} + \cdots + X_{ip}$ . Splitting the set of indices $J = 1,...,d$ in disjoint subsets $I_1,...,I_m$ can be
representation as:
$$X = X_{I_1} + \cdots + X_{I_m} \qquad \text{(III)}$$

The eigentriple grouping is the process of choosing the sets $I_1,...,I_m$. Finally diagonally
averaging transfers each I matrix into a time series, which is an additive component of the
initial series $Y_T$. If $z_{ij}$ is an element of the matrix $Z$. Then k-th term of the produced series is
acquired by averaging $z_{ij}$ for $i$, $j$ such that $i + j = k + 2$. This procedure is known as diagonal
averaging (Hankelization) of the matrix $Z$. The Hankelization of a matrix $Z$ is the Hankel matrix
$HZ$ which is the trajectory matrix corresponding to the series obtained from diagonal averaging.
Hankelization is a best logical technique that $HZ$ matrix is the nearest to $Z$ among all





corresponding size of Hankel matrices (Golyandina et al., 2001). Now applying the
Hankelization technique to each and every matrix components in equation III, we get
$$X = \tilde{X}_{I_1} + \ldots + \tilde{X}_{I_m} \qquad \text{(IV)}$$

Here $X_{I1} = HX$ corresponding to initial series $Y_T = (y_1, \ldots, y_T)$ decomposition into a
sum of m series as
$$y_t = \sum_{k=1}^{m} y_t^{\sim(k)} \qquad \text{(V)}$$

Here $y_t^{\sim(k)} = y_1^{\sim(k)} + \ldots + y_T^{\sim(k)}$ corresponds matrix $X_{Ij}$.
**4      Results**
The average value of radon for a period of six month from April 2017-September 2017
was found to be in the range 55-117 kBq/m$^3$. The average emanation of soil-gas radon at
MPGO for April, May, June, July, August and September 2017 is reported to be 55.94, 93.11,
109.12, 117.69, 101.45, 92.34 (kBq/m3) respectively with standard deviation (Std.) of 21.3,
28.53, 19.07, 28.09, 25.86, 18.65 (kBq /m3) respectively. Simultaneously variation of soil
temperature and pressure with radon emanation was observed. Usually, radon shows positive
correlation with temperature i.e. the soil radon concentration increases with increase in
temperature and decreases as temperature decrease. The correlation coefficient (Pearson
correlation) between radon and temperature is found to be 0.5 signifying positive correlation,
while the correlation coefficient between radon and pressure is found to be -0.5 signifying
negative correlation with an average temperature and pressure of 28.60 $^0$C and 991.03 mbar
respectively during afore mentioned period. The positive correlation of radon with temperature
might be due to the rise in diffusion rate with temperature (Sharma et al. 2000, Singh et al.
2008). The negative correlation coefficient was found for soil radon and pressure which signify,
with the increase in pressure, the radon emanation decreases while with the decrease in pressure




the radon emanation increases. In general, the negative correlation is due to the diffusion
process which slows down with increase in pressure, which in turn decreases the radon
concentration in the soil. The average value of pressure, temperature, standard deviation,
percentage (%) correlation coefficient for the observation period is detailed in Table 1. The
maximum and minimum temperature observed was 31.24 $^0$C and 23.78 $^0$C i.e. a change of 7.46
$^0$C during the period of observation. Simultaneously, the maximum and minimum pressure
during the period of observation was 999.32 mbar and 980.72 mbar i.e. a change of 18.6 mbar.
Digital filter (Butterworth) is applied to eliminate the high frequency quasi periodic
components form the soil radon time series for better discernibility of seismic induced
anomalies and is represented in Figure 2.

204       The covariance matrix of the first 9 group of soil radon (Rn-222) time series is

represented in Figure 3.The singular value decomposition (SVD) to Rn-222 data evinced that
first 9 eigenfunctions (Figure 4) when grouped resulted for 99.90 % of the total variance in the
individual time series. The eigenfuction group 1 and 2 represents the aperiodic component and
group 3 to 9 represented periodic components. The periodic and aperiodic component mostly
corresponds to diurnal and semidiurnal variation (Kumar et al., 2015). The decomposed
eigenvectors in soil radon time series is grouped into two classes as diurnal and semidiurnal
variation. The sum of eigenfuction group 1 and 2 accounts for 98.62 % and group 3 to 9
accounts for 0.48 % of the variance. Radon variations is governed by daily variations, which
accounts to 99.90 % of the total variance in soil gas radon at the MPGO, Tezpur. The Principal
Component of soil radon (Rn-222) related to the first 9 grouping of eigentriples is represented
in Figure 5 and w-correlation matrix for the 9 reconstructed components is represented in
Figure 6.

217       The covariance matrix of the first 9 group of soil temperature and pressure time series

is represented in Figure 7 and Figure 11 respectively. The decomposed eigenfunctions for soil

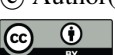



temperature and pressure time series by applying SSA is represented in Figure 8 and Figure 12
respectively. The first 9 eigenfunction group from SVD to atmospheric temperature records
accounts nearly about 99.99 % and first 9 eigenfunction group of soil pressure 100 % of the
variation respectively. It is discernible that first eigenfunction 1 alone itself is capable of
producing 100 % variation but the time series is well modeled when first 9 eigenfunction is
grouped.  This evinces SVD to radon, soil temperature and soil pressure data evince that first
9 eigenfunction accounts >98.00 % of the variance of the individual data series. This is
fascinating to observe that sum of first 9 eigenfunction is fairly sufficient to reproduce the
prominent features of the overall variation. This also suggests that the most significantly
produced variations are mostly free from naturally induced variations. Soil pressure variations
are dominated by semidiurnal variations by 100 % of the total variation in atmospheric pressure
at MPGO, Tezpur. On the other hand soil temperature variations is dominated by daily
variations like radon variation which account to 97.99 % whereas soil pressure accounts for
100 % of the total variance. This suggest that daily variations of soil radon (Rn-222) emanation
are controlled by soil pressure in MPGO, Tezpur during the investigation period followed by
soil temperature. The Principal Component of soil temperature related to the first 9 grouping
of eigentriples is represented in Figure 9 and w-correlation matrix for the 9 reconstructed
components is represented in Figure 10. The Principal Component of soil pressure related to
the first 9 grouping of eigentriples is represented in Figure 13 and w-correlation matrix for the
9 reconstructed components is represented in Figure 14. The reconstructed time series for soil
radon (Rn-222), temperature and pressure is represented is Figure 15 and it's residual in Figure
16. It is also discernible that the during the investigation time period the  pressure change was
more than temperature change which also evinces the variation of soil radon at MPGO, Tezpur
was more governed by pressure followed by temperature change.  The quasi periodic, diurnal
(periodic mostly due to temperature) and semidiurnal (aperiodic mostly due to pressure) were





eliminated by Singular Spectrum Analysis in the reconstructed time series by grouping and
analyzing the eigenfunctions and principle component of individual time series of Rn-222,
temperature and pressure respectively. The soil pressure and temperature were found to be
negatively correlated to each other (-0.62) which produces a pseudo effect in the soil radon
time series. The grouping and reconstruction of the time series also eliminates these pseudo
effect arising due soil pressure and temperature.

**5      Discussion**

251         The reconstructed soil radon time series along with the seismic activity during the

investigation period is shown in Figure 17. The hypocentral parameters of the earthquake
events found to have correlation with soil radon emanation is listed in Table 2. It was observed
that there were certain positive amplitude rise anomalies in radon emanation prior to six out of
9 earthquakes which occurred in the vicinity (100 km radially from MPGO,Tezpur). Increase
in the soil radon concentration is generally assigned to developments of microcracks, fractures
and fissure caused by dilatancy prior to earthquake. This process enhances the transportation
of radon from its original enclosure following the cracks. The rise in soil radon concentration
prior to an earthquake may be due to the strain buildup processes in the area. During this
process, very small fractures are developed in the rocks which enhances the contribution of
radon gas to the soil near the surface of earth (Miklavčić et al., 2008). Three earthquake events
were preceded by negative anomalies. The negative anomalies might be due to the
circumstance that during the final stage of dilatancy model prior to an earthquake the Rn-222
emanation can be stable or it can decrease (Tomer, 2016). This is because, during the final
stage prior an earthquake, rupture occurs and fluid pressure and stress on rocks is released
(Bakhmutov and Groza, 2008). The fluid pressure increases resulting in water level rise and
this does not allow the soil gas Rn-222 to escape from the surface which in turn reduces or
stabilizes the emanation of Rn-222. Further a decreasing radon anomaly as observed in this





study may be the result of squeezing effect of compressional stress built up in the rock, which
in turn result in soil porosity changes at micro scale. There were certain events which occurred
on the same day or just a very short seismic gap of 1 or 2 days. Here in the case earthquake
with higher magnitude might also be the probable reason for the anomalous behavior of the
soil gas radon emanation, as for spatio-temporal clustered earthquakes, the largest magnitude
earthquake is presumed to precede the anomalies in radon emanation (Hartmann and Leavy,
2005).  Positive as well as negative anomalies were observed prior to 9 events which occurred
in the vicinity (100 km radially from MPGO) with in a short span of time.
**6      Conclusion**

278         The investigation concludes that digital filter assists in eliminating the high frequency

quasi periodic components from the time series. The SSA method helps eliminating the diurnal
and semidiurnal fluctuations from soil radon time series for improved correlating and detection
of the soil radon emanation with seismic activity. The investigation also evinced that radon is
dominated by daily variation at MPGO,Tezpur and is controlled by soil pressure followed by
temperature. It is also concluded that principle component analysis helps in removing the
pseudo effect pertaining to simultaneous soil pressure and temperature effect. It was observed
that there were certain positive amplitude rise anomalies in radon emanation prior to six events
out of 9 earthquakes which occurred in the vicinity (100 km radially from MPGO,Tezpur)
within a short span of time. The increase of radon emanation with temperature might be the
result of increasing diffusion rate with temperature. Three earthquake events were preceded by
negative anomalies. The negative anomalies might be due to the circumstance that during the
final stage of dilatency model prior to an earthquake the soil gas radon emanation can be stable
or it can decrease. This is because, during the final stage prior an earthquake, rupture occurs
and fluid pressure and stress on rocks is released. Further a decreasing radon anomaly as





observed in this study may be the result of squeezing effect of compressional stress built up in
rock, which in turn changes porosity of soil at micro scale.
***Acknowledgements***. We acknowledge our sincere thanks to Ministry of Earth Sciences
(MoES) Government of India for providing funds vide project no.MoES/P.O.(Seismo)/NPEP-
16/2011. We thank Director, CSIR-NEIST Jorhat for giving necessary permission to publish
this paper.

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

**FIGURES AND TABLES**

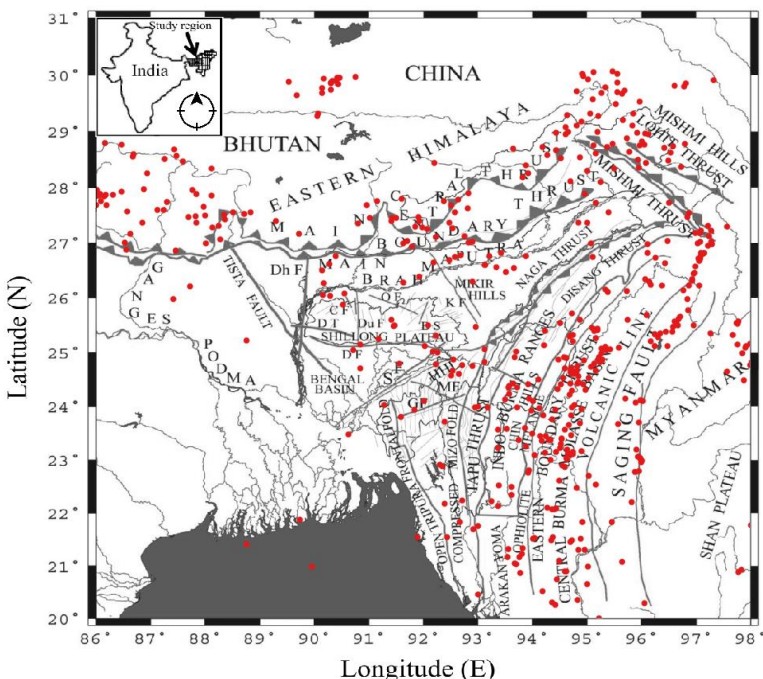

**Figure 1:** Map illustrates the earthquake events of Mw $\geq$ 5 during 1918 to 2018, in NE-India

and its border region ($20^0$-$30^0$ N and $86^0$-$98^0$ E) along with the major tectonic features of the

region.


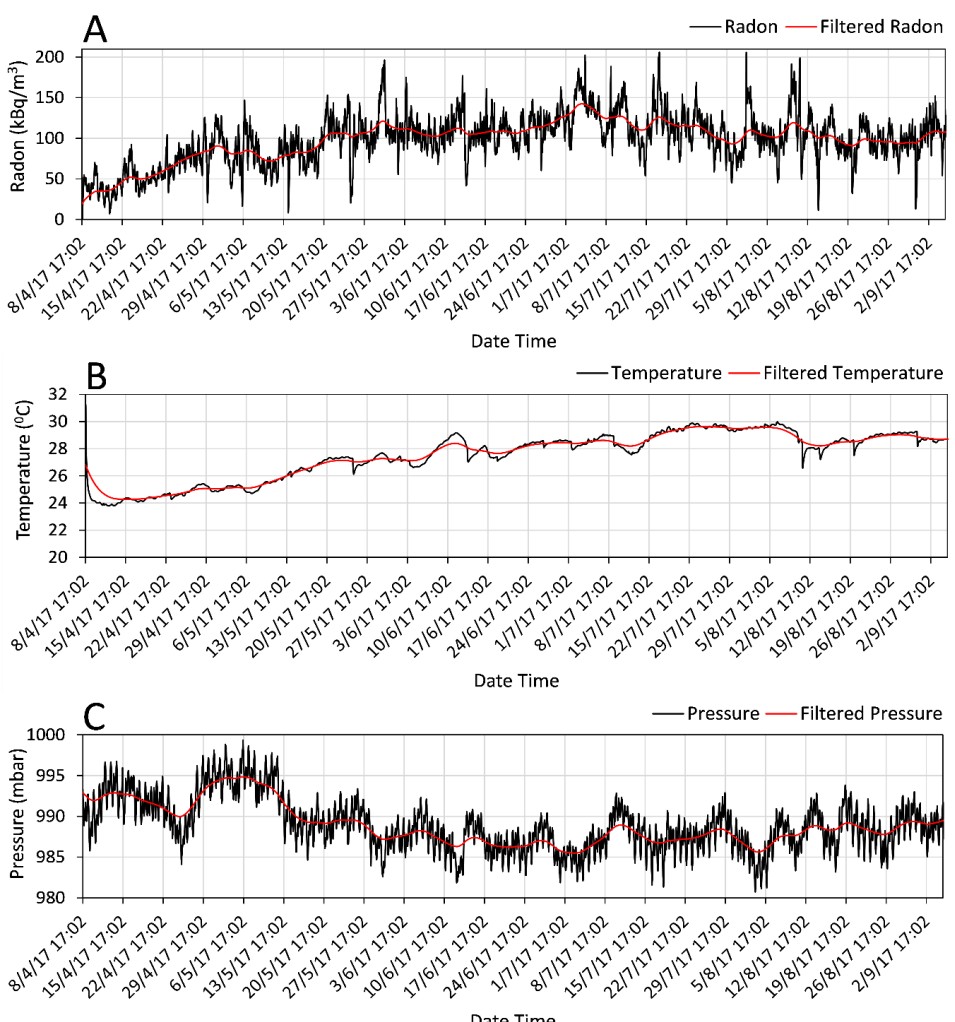


**Figure 2:** The plot represents the removal of high frequency quasi periodic component for A)

filtered time series of soil radon, B) filtered time series of soil temperature and C) filtered time

series of soil pressure.




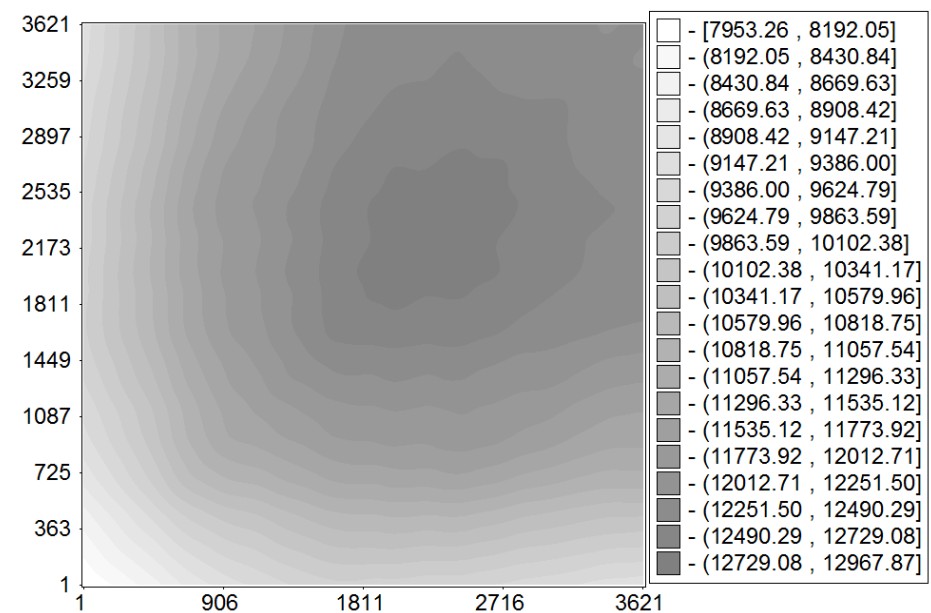

**Figure 3:** Covariance matrix of the first 9 group of soil radon (Rn-222) time series.


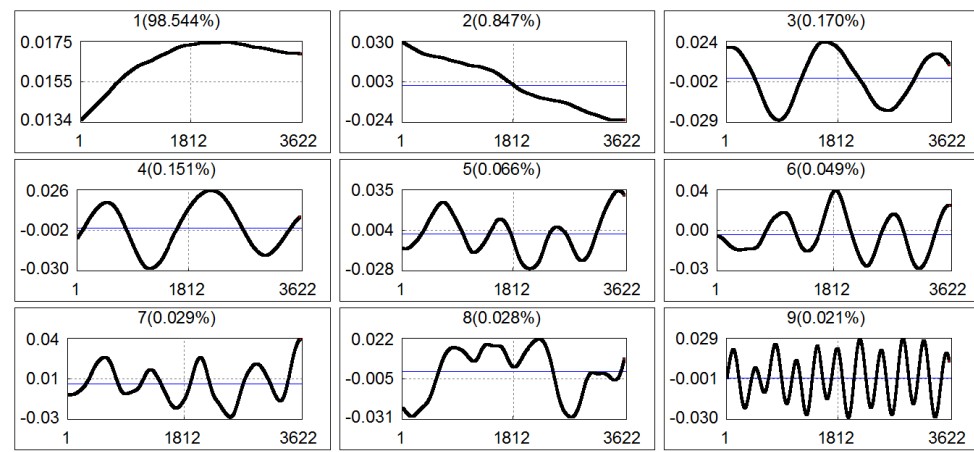


**Figure 4:** Eigenfunctions of soil radon (Rn-222) first 9 group.




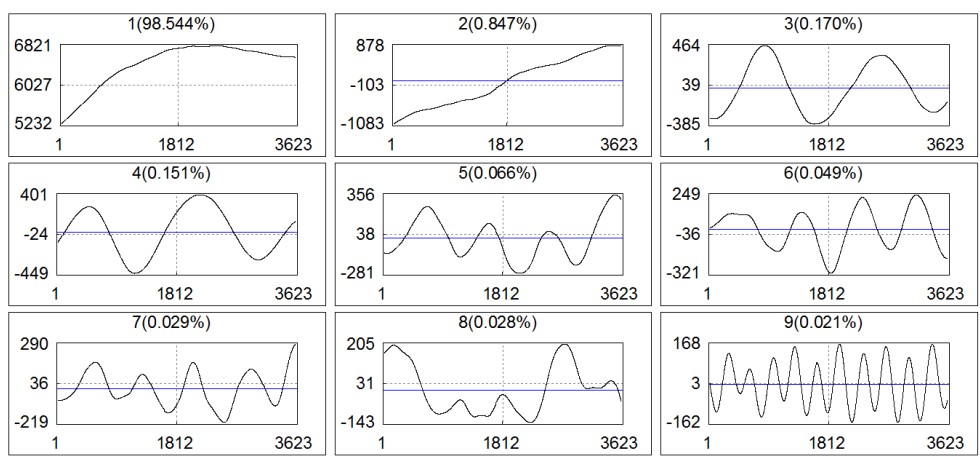


**Figure 5:** Principal Component of soil radon (Rn-222) related to the first 9 eigentriples.


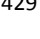

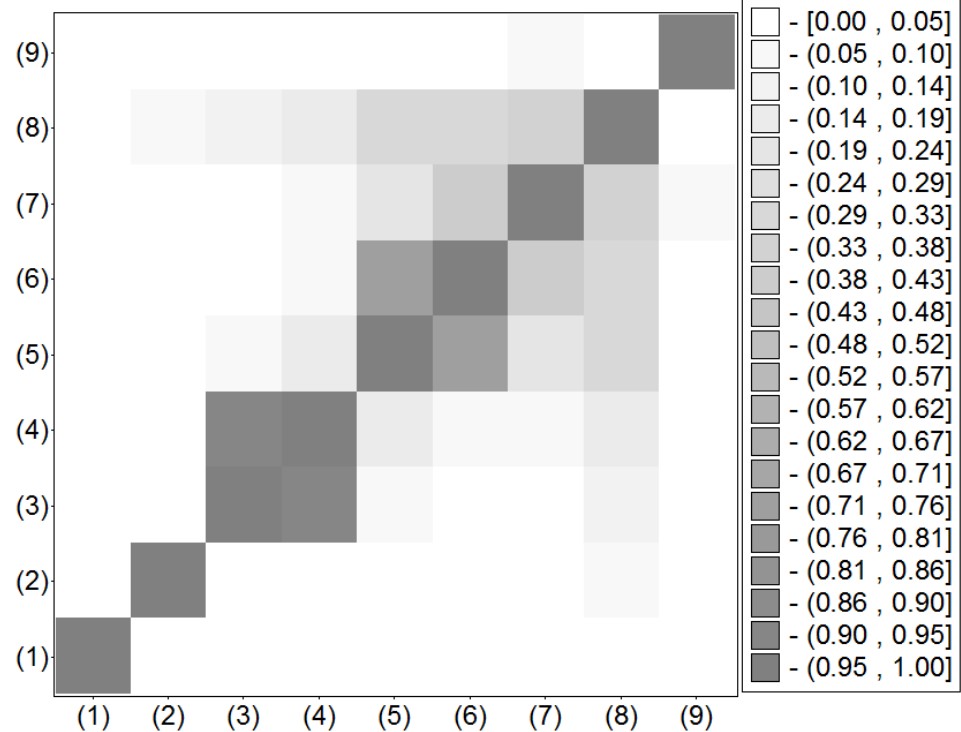


**Figure 6:** w-correlation matrix for the 9 reconstructed components of soil radon time series.





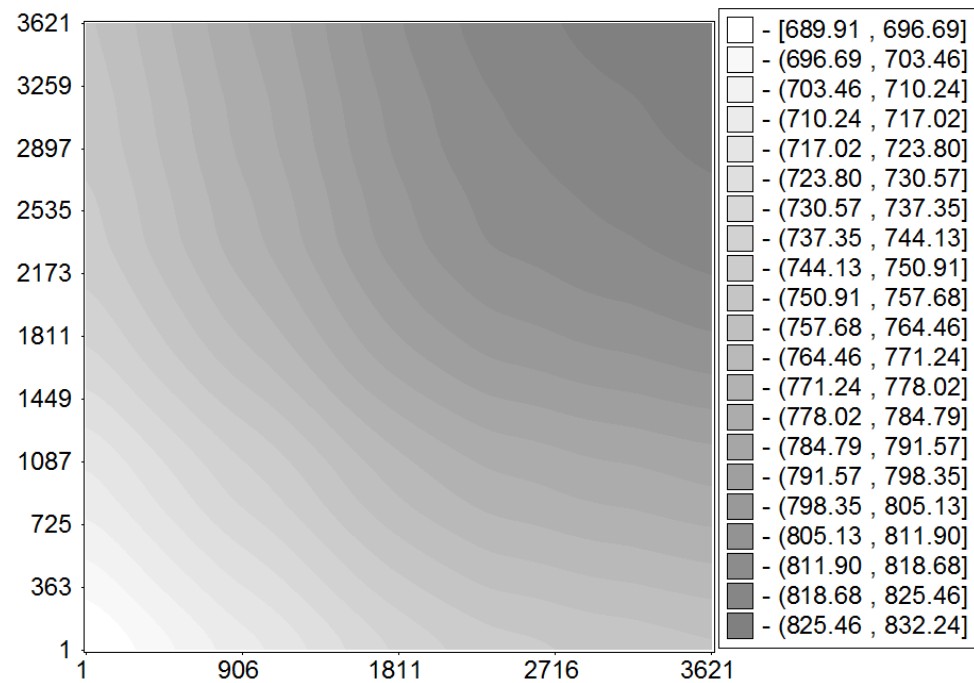


**Figure 7:** Covariance matrix of the first 9 group of soil temperature time series.


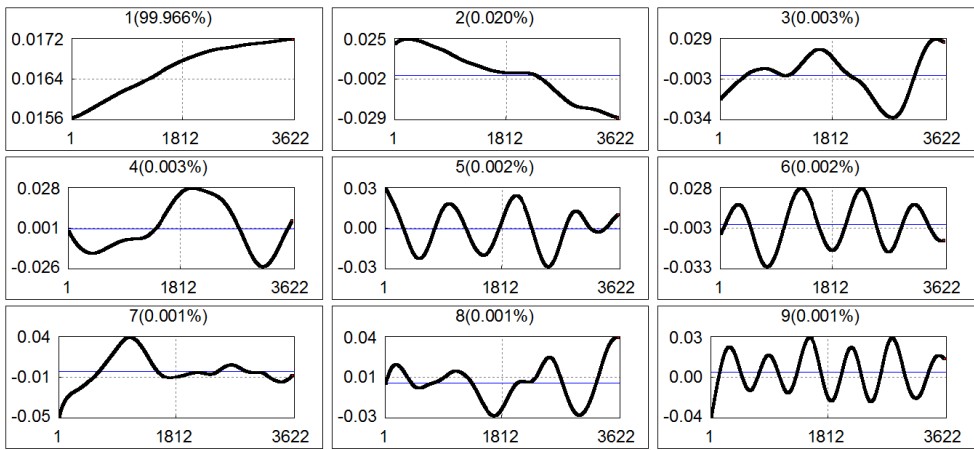


**Figure 8:** Eigenfunctions of soil temperature first 9 group.

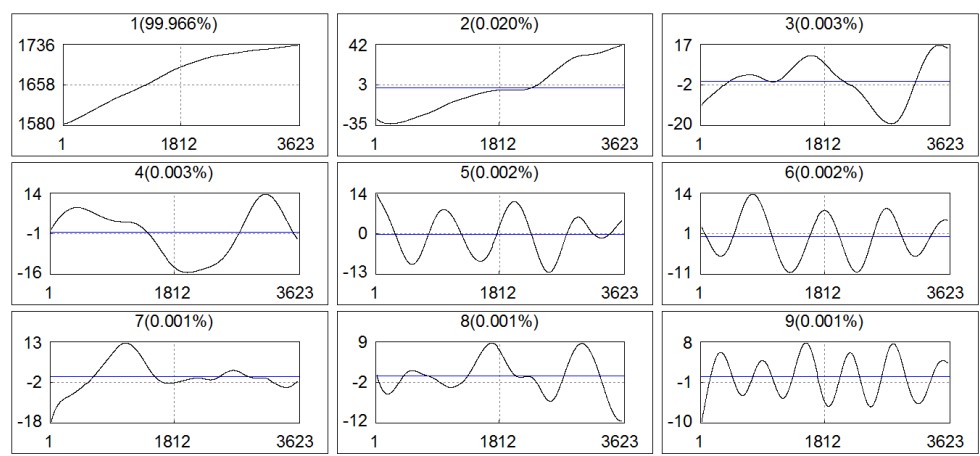

**Figure 9:** Principal Component of soil temperature related to the first 9 eigentriples.

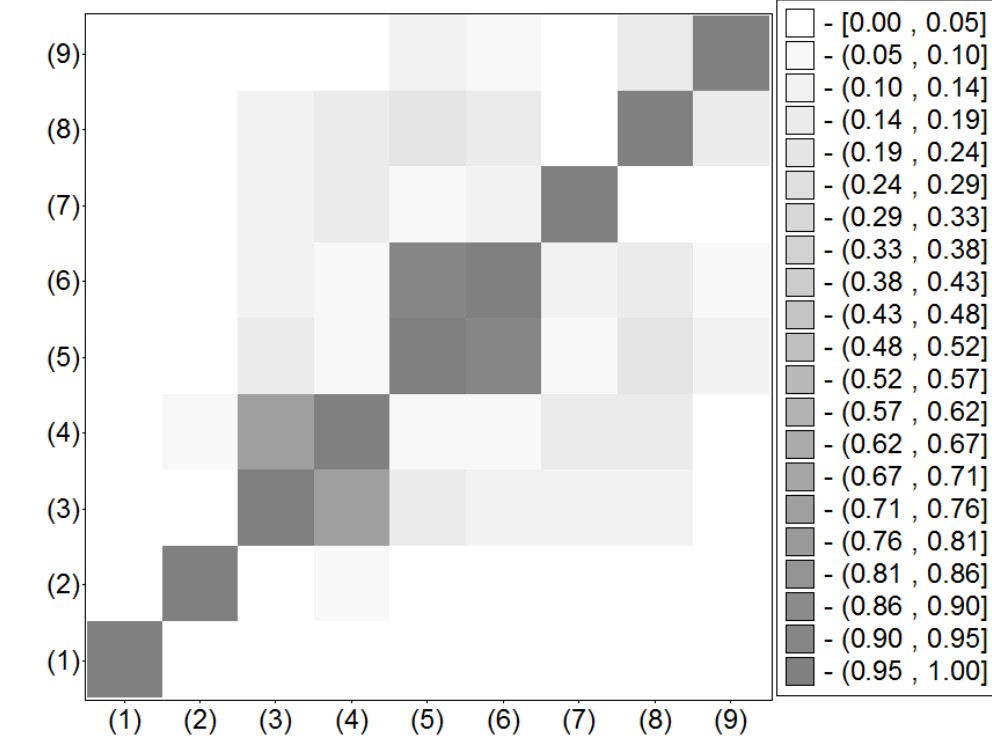

**Figure 10:** w-Correlation matrix for the 9 reconstructed components of soil temperature time series.


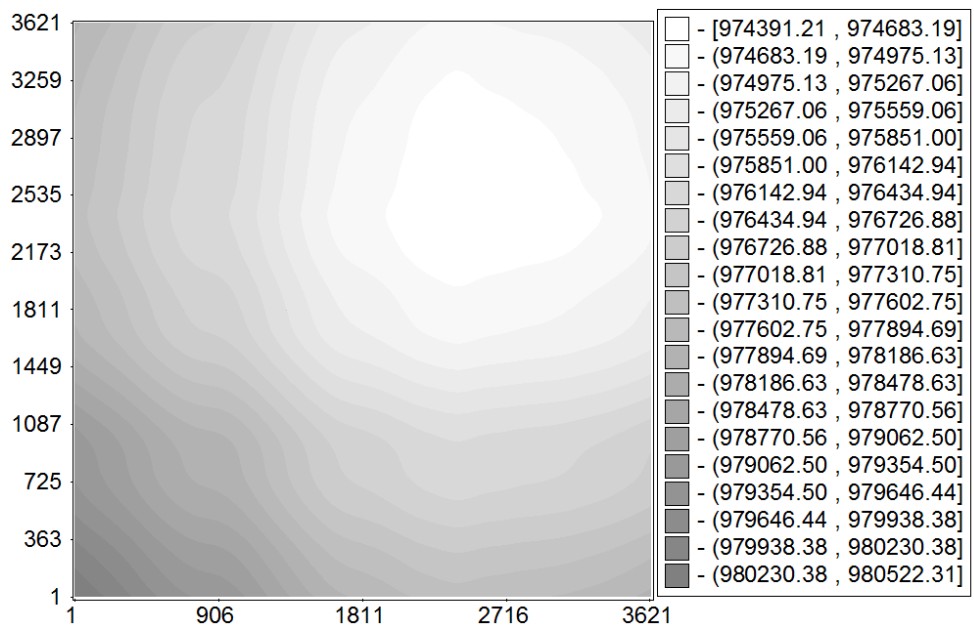


**Figure 11:** Covariance matrix of the first 9 group of soil pressure time series

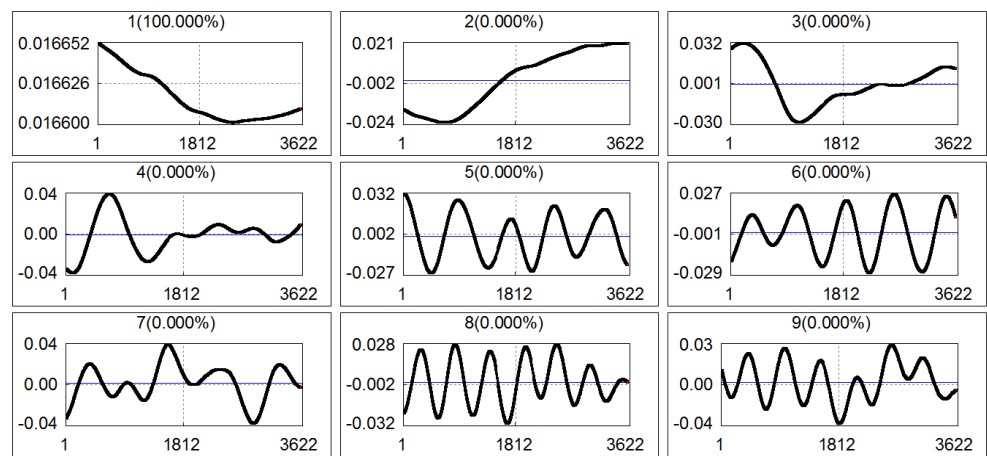


**Figure 12:** Eigenfunctions of soil pressure first 9 group.




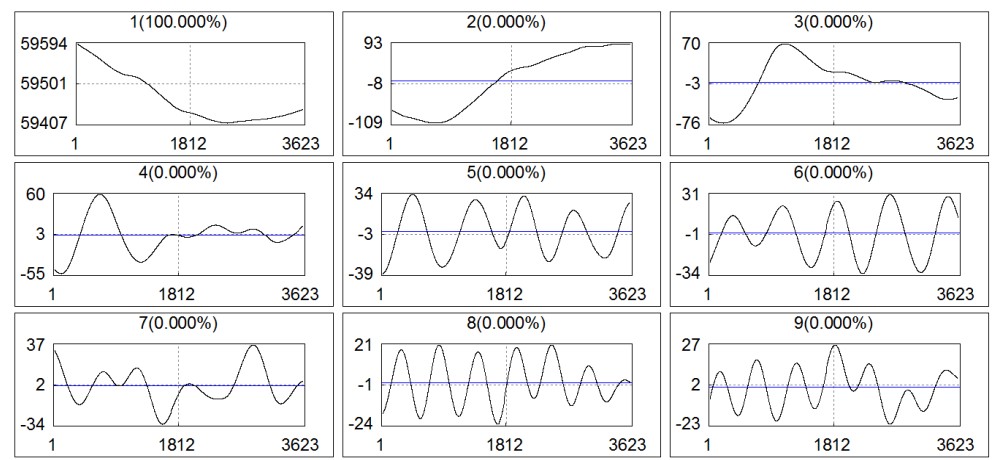


**Figure 13:** of soil temperature related to the first 9 Eigentriples.

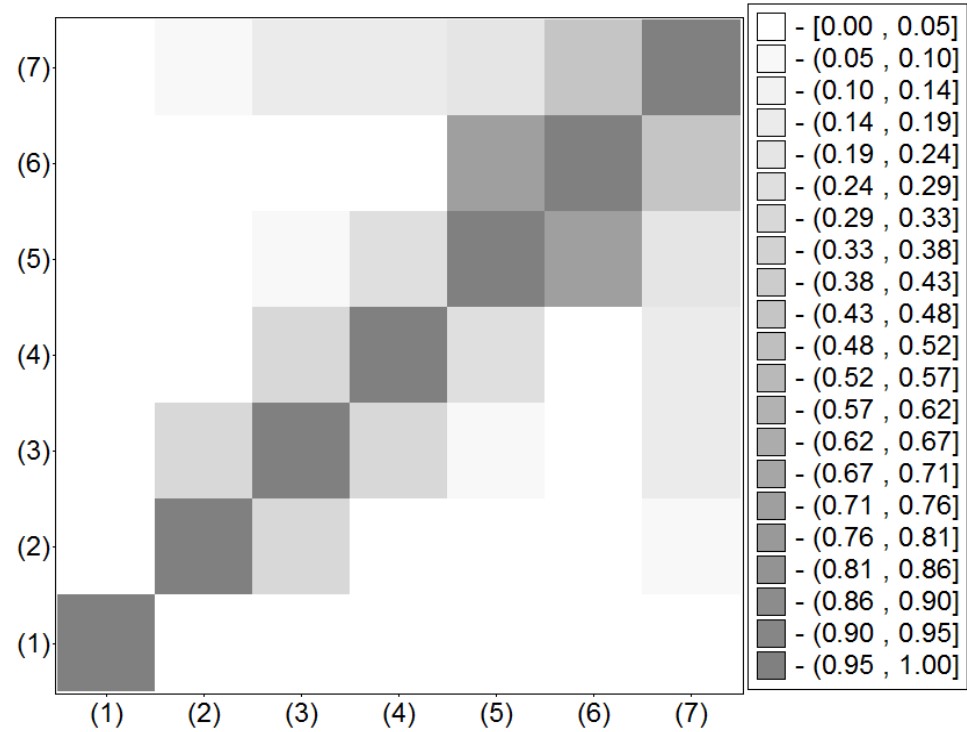


Figure 14: w correlation matrix for the 9 reconstructed components of soil pressure time series.





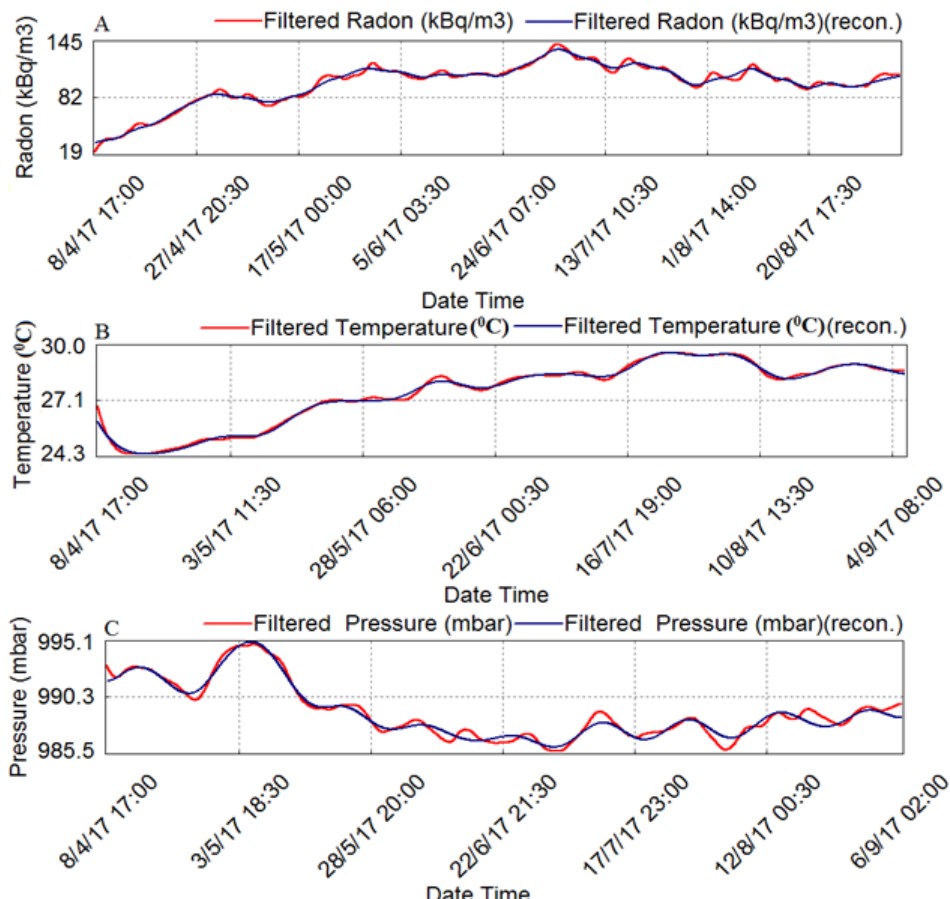


**Figure 15:** Plot showing the reconstructed time series of A) filtered soil radon, B) filtered temperature and C) filtered pressure.








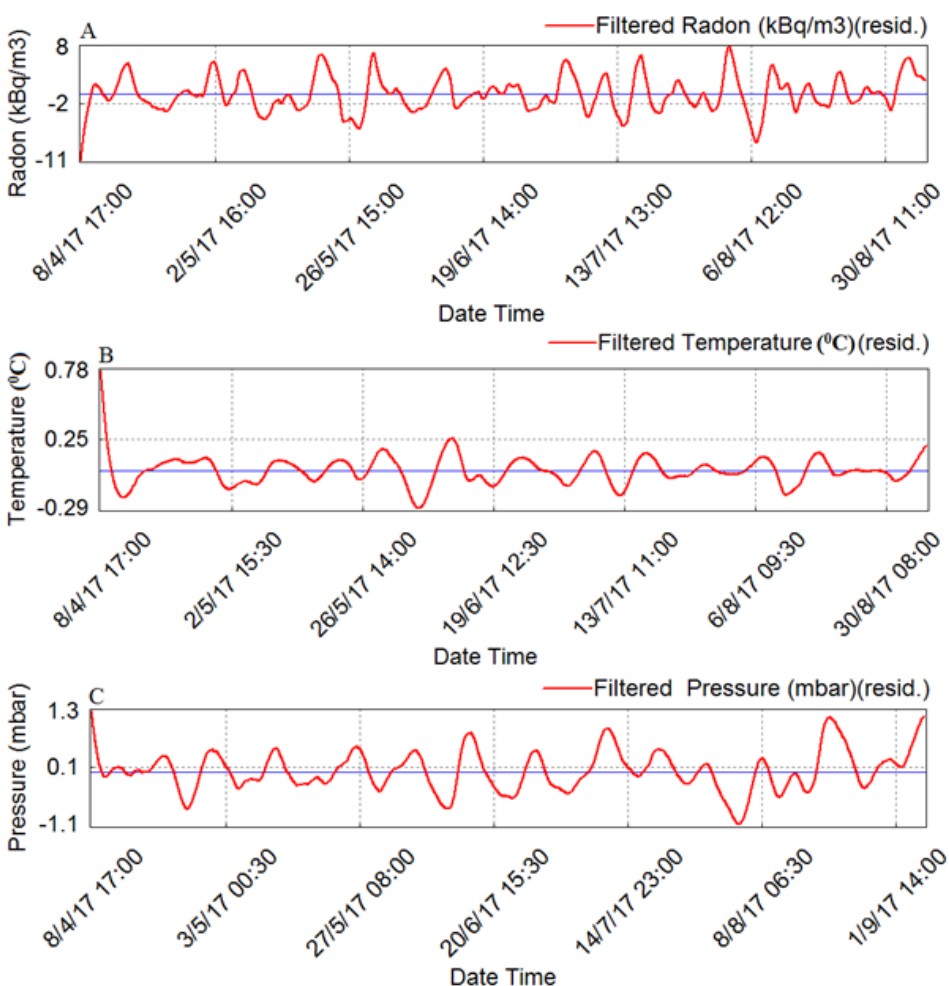


**Figure 16:** Residual of reconstructed time series of A) Filtered soil radon, B) temperature and

C) pressure respectively.




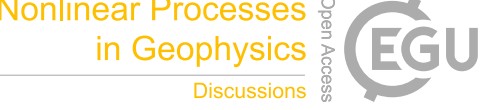
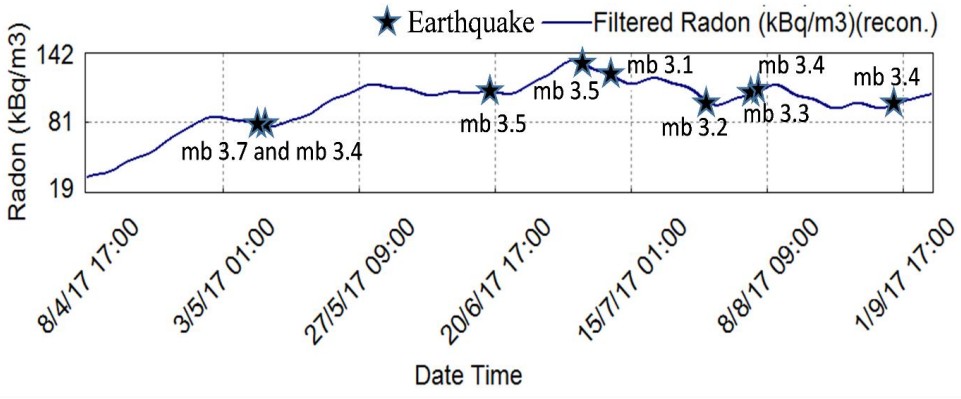


**Figure 17:** Plot showing the reconstructed filtered time series of soil radon emanation along with earthquake during the investigation period in the vicinity of MPGO, Tezpur (100 km radially from MPGO) which occurred in a very short span of time.
























**Table 1:** The correlation co-efficient of soil radon gas concentration with soil pressure and
temperature at OH-MPGO during year 2017.

| Parameters | Average (Avg.) | Standard Deviation (Std.) | % Coefficient (Std. /Avg.) | Variation | Correlation Coefficient |
|---|---|---|---|---|---|
| **Radon (KBq/m³)** | 94.94 | 23.58 | 24.84 | | ---- |
| **Temperature (⁰C)** | 28.60 | 0.62 | 2.19 | | 0.5 |
| **Pressure (mbar)** | 991.03 | 2.48 | 0.25 | | -0.5 |






















**Table 2:** Hypocentral parameters of the earthquake events found to have correlation with radon
anomaly.

| Date of Event | UTC TIME | Lat (°N) | Long (°E) | Place | Depth (km) | Mag (m$_b$) | Distance from MPGO (km) |
|---|---|---|---|---|---|---|---|
| **09/05/2017** | 01:53:55 | 26.3 | 92.7 | Assam | 25 | 3.7 | 44 |
| **09/05/2017** | 03:26:54 | 26.6 | 93.2 | Assam | 28 | 3.4 | 67 |
| **20/06/2017** | 04:31:58 | 27.1 | 92.5 | West Kameng, Arunachal Pradesh | 10 | 3.5 | 67 |
| **04/07/2017** | 10:05:47 | 27.0 | 92.1 | West Kameng, Arunachal Pradesh | 10 | 3.5 | 78 |
| **10/07/2017** | 23:28:30 | 27.1 | 93.8 | Papumpare, Arunachal Pradesh | 10 | 3.1 | 78 |
| **25/07/2017** | 18:28:00 | 26.3 | 93.1 | Karbi Anglong, Assam | 28 | 3.2 | 67 |
| **05/08/2017** | 12:24:56 | 26.8 | 92.2 | Darrang, Assam | 10 | 3.3 | 44 |
| **07/08/2017** | 11:25:07 | 26.3 | 91.7 | Kamrup, Assam | 10 | 3.4 | 100 |
| **31/08/2017** | 17:57:26 | 26.6 | 92.7 | Sonitpur Assam | 10 | 3.4 | 67 |





