# Peer review of "Singular spectrum and principal component analysis of soil radon"

_Nonlinear Processes in Geophysics, 2019_

## Referee Comment (RC1) · Anonymous Referee #1 · 21 Aug 2019

General comments This study is about anomalies in radon concentration related to earthquakes using singular spectrum analysis, model free method. More detail explanation is needed, since overall there is a lack of explanation of analysis. There are too many figures showing the results. Please provide as many figures as you need to support the results. It is necessary to explain the relevance and differences of previous studies using singular spectrum analysis for radon concentration data.

Specific comments 1. The way to calculate the covariance matrix showed in Fig. 3, 7 and 11 should be described. Scientific implication and/or explanation should be

included, since few descriptions regarding the figures are in the present paper.

2. Eigenfunctions in Fig. 4, 8, 12 and principle components in Fig. 5, 9, 13 are almost the same considering the difference in sign. Detail explanations how to calculate them are needed in method section.

3. Since the data used in the result section are filtered ones, diurnal and semidiurnal components seemed to be not included in the data. What is the basis of the description regarding lines 20-23 and lines 228-230?

4. Regarding line 63, the reason of criteria, 100 km of mb > 3.1, for selecting seismic activity should be explained.

5. Regarding lines 158-163, lack of description of the way to group the elementary matrices.

6. Regarding lines 186-190 and Table 1, Is which used to calculate correlation coefficient, original data or filtered data? Are the results just 0.5 and -0.5? If not, the smaller digit values should be indicated in Table 1 (e.g. 0.49)

7. Explanation about the w-correlation matrix is needed at method section.

8. Regarding lines 240-242, It can be suggested that the pressure with larger change is dominant, only when the response of radon to temperature and pressure is equal. However, no evidence the response of radon to them are indicated in the paper. Therefore, there is lack of basis for this description.

9. Regarding lines 253-255, detail descriptions indicating which earthquakes have positive anomaly and the others have negative anomaly should be added. Are these results corrected the effects of temperature and pressure? Otherwise, it cannot be distinguished whether it is a change due to an earthquake or a change due to temperature or pressure.

10. Regarding lines 266-268, Raising the water level means that there is a pressure

gradient, which means that the fluid flows from a place with high pressure to a place with low pressure. Radon can be thought of as moving through the ground as well, and rising water levels can also indicate an increase in radon concentration. Please indicate if there is any previous research that supports the argument, in lines 266-268, in the paper.

Technical comments 1. Line 14, the complete name of MPGO also should be indicated here. 2. Lines 52-53, "Latitude 26.61o; Longitude 92.78o" should be "Latitude N26.61o; Longitude E92.78o". 3. Line 64, "The major problem arises is the" should be "The major problem is the". 4. Line 68, add spaces like "Stranden et al., 1984; Kumar et al., 2009; Walia et al., 2005". 5. Line 72, "parameters on radon emanation" should be "parameters on radon concentration". 6. Line 150, add brackets like "i.e. (Ui, Uj) = 0 for". 7. Lines 182, and 183, "m3" should be "m3". 8. Line 197, does percentage correlation coefficient mean coefficient of variation? 9. Fig. 4, 5, 8, 9, 12 and 13, need axis label. 10. "0C" should be "oC". 11. Line 408, "(200-300 N and 860-980 E)" should be "(20o -30o N and 86o -98o E)".

---

## Author Comment (AC1) · 27 Aug 2019

We are very grateful and convey our sincere thank you to your generous comments and significant suggestion for the manuscript. We have revised the manuscript to best of our knowledge as per your comments and suggestions. We have submitted our response in the "npg-2019-37-supplement.zip" folder as a supplement which contains pdf documents of the following: 1. Reply to Anonymous Referee #1 2. REVISED MANUSCRIPT WITH TRACK CHANGE 3. REVISED MANUSCRIPT (Without track change)

Please also note the supplement to this comment:
https://www.nonlin-processes-geophys-discuss.net/npg-2019-37/npg-2019-37-AC1-supplement.zip

---

## Referee Comment (RC2) · Anonymous Referee #1 · 10 Sep 2019

General comments

I would like to thank authors for trying to address my concerns, although they mostly remain. The manuscript needs further clarifications and improvements before I can recommend it for publishing. Please see below.

Specific comments

1. The equation showed at (VI) is wrong. The equation indicates autocovariance not about covariance. Please show the correct an equation used in the study.

2. The equation (VII) is same to the equation (VI). Is S a matrix? The equation does not show the size of the matrix and the element, i-th row and j-th column. Please show the correct an equation used in the study.

3. Regarding lines255-264. Kumar et al. 2015 does not claim "The periodic and aperiodic component mostly corresponds to diurnal and semidiurnal variation.". They claim that periodic components of radon data extracted by SSA consist of diurnal and semidiurnal components, since the components includes variation of 24-hour and 12-hour period. Therefore, lines 255-264 is not correct.

4. Regarding Fig. 4, 5, 8, 9, 12 and 13. The period in the eigenfunction components 3-9 (Fig. 4) is not clear, since no information about the unit of x-axis of the figure. The other figures 5, 8, 9, 12 and 13 have same problems.

Technical comments 1. Line 187. The equation is not correct.

---

## Referee Comment (RC3) · Anonymous Referee #2 · 24 Sep 2019

The paper doesn't present new results and methodologies. The correlation between radon and seismicity, that seems to be the goal of the paper, is treated in a very poor way, and the results seem to be mixed among the various earthquakes.

More specifically:

- Singular Spectrum Analysis: this chapter needs absolutely an explanation more detailed and a check about formulas.

- Results: line 179..from April to September... Add a little table instead of putting the

monthly average radon values in the text. line 204: Explain why the groups are 9.

- Discussion: it's difficult to understand to which event is related each explanation. You write about close events, even a few hours or days of difference and that theoretically the radon anomaly is linked to the major. I think that one purpose of this article could be precisely to try to identify possible time and space windows, which make it possible to understand if radon anomalies are possibly linked to a single event or if they are the cumulative effect of several earthquakes; it is sort of declustering as it's done in the probabilistic seismic hazard computations.

- References: must be re-checked: some references are missing and for some there is a difference between the text and the references (different number of authors, wrong year of publication ...)

- Figures: - 1 it would be useful to add a window showing the study area of 100 km around Tezpur, and the 9 earthquakes studied; - 3 to 14 more explanations in the captions were necessary, but I see you already add them in the corrected version; - maybe it is possible to accorpate some figures (the matrix ones...for example...).

Minor corrections:

100 earethquakes...correct as earthquakes 136 the common most algoritm....correct as...the most common algoritm; 202 form.....correct as....from

---

## Author Comment (AC2) · 2 Oct 2019

We are very grateful and convey our sincere thanks for your generous comments and significant suggestion for the manuscript. We have revised the manuscript to best of our knowledge as per your comments and suggestions. We have submitted our response in the "npg-2019-37-supplement.zip" folder as a supplement which contains pdf documents of the following: (1) Reply 2 to Anonyous Refree#1 (2) REVISED MANUSCRIPT WITH TRACK CHANGE (3) REVISED MANUSCRIPT (Without track change)

[Figure]

Please also note the supplement to this comment:
https://www.nonlin-processes-geophys-discuss.net/npg-2019-37/npg-2019-37-AC2-supplement.zip